# Molecular Dynamics Mappings of the CCT/TRiC Complex-Mediated Protein Folding Cycle Using Diffracted X-ray Tracking

**DOI:** 10.3390/ijms241914850

**Published:** 2023-10-03

**Authors:** Kazutaka Araki, Takahiro Watanabe-Nakayama, Daisuke Sasaki, Yuji C. Sasaki, Kazuhiro Mio

**Affiliations:** 1AIST-UTokyo Advanced Operando-Measurement Technology Open Innovation Laboratory (OPERANDO-OIL), National Institute of Advanced Industrial Science and Technology (AIST), 6-2-3 Kashiwanoha, Chiba 277-0882, Japan; kazu.mio@aist.go.jp; 2WPI Nano Life Science Institute (WPI-NanoLSI), Kanazawa University, Kakuma-machi, Kanazawa 920-1192, Japan; tnakawata@se.kanazawa-u.ac.jp; 3Graduate School of Frontier Sciences, The University of Tokyo, 5-1-5 Kashiwanoha, Chiba 277-8561, Japanycsasaki@edu.k.u-tokyo.ac.jp (Y.C.S.)

**Keywords:** chaperonin, single molecular dynamics, diffracted X-ray tracking

## Abstract

The CCT/TRiC complex is a type II chaperonin that undergoes ATP-driven conformational changes during its functional cycle. Structural studies have provided valuable insights into the mechanism of this process, but real-time dynamics analyses of mammalian type II chaperonins are still scarce. We used diffracted X-ray tracking (DXT) to investigate the intramolecular dynamics of the CCT complex. We focused on three surface-exposed loop regions of the CCT1 subunit: the loop regions of the equatorial domain (E domain), the E and intermediate domain (I domain) juncture near the ATP-binding region, and the apical domain (A domain). Our results showed that the CCT1 subunit predominantly displayed rotational motion, with larger mean square displacement (MSD) values for twist (χ) angles compared with tilt (θ) angles. Nucleotide binding had a significant impact on the dynamics. In the absence of nucleotides, the region between the E and I domain juncture could act as a pivotal axis, allowing for greater motion of the E domain and A domain. In the presence of nucleotides, the nucleotides could wedge into the ATP-binding region, weakening the role of the region between the E and I domain juncture as the rotational axis and causing the CCT complex to adopt a more compact structure. This led to less expanded MSD curves for the E domain and A domain compared with nucleotide-absent conditions. This change may help to stabilize the functional conformation during substrate binding. This study is the first to use DXT to probe the real-time molecular dynamics of mammalian type II chaperonins at the millisecond level. Our findings provide new insights into the complex dynamics of chaperonins and their role in the functional folding cycle.

## 1. Introduction

The molecular chaperone CCT/TRiC complex also known as chaperonin containing TCP-1, herein referred to as the CCT complex, belongs to the type II chaperonin family and is involved in cellular protein folding, degradation, and quality control [1,2,3]. The CCT complex, present in all eukaryotic cells, consists of eight distinct but homologous subunits, namely CCT1-α(A), CCT2-β(B), CCT3-γ(G), CCT4-δ(D), CCT5-ε(E), CCT6-ζ(Z), CCT7-η(H), and CCT8-θ(Q) (Figure 1A). It also plays a vital role in maintaining the structure and function of the cytoskeleton, facilitating the folding of actin and tubulin [4,5,6,7].

The ring arrangement of the CCT complex imparts asymmetric functionality [10,11] (Figure 1B). On one side, CCT7/8/6/3 subunits with low ATP binding affinity form a positively charged inner face. On the other side, CCT1/4/2/5 subunits exhibit high ATP binding, forming a negatively charged inner face [12,13]. Each subunit architecture of the CCT complex includes three domains: an equatorial domain (E domain) with the ATP binding regions, an apical domain (A domain) presumably involved in unfolded substrate binding, and an intermediate domain (I domain) connecting the E and A domains [14] (Figure 1C,D).

During its functional cycle, the CCT complex undergoes conformational rearrangements driven by ATP binding and hydrolysis. These rearrangements enable the complex to transition from an open to a closed state, creating a folding cavity for client proteins. A previous molecular simulation study demonstrated that conformational changes were accompanied by the dynamic and asymmetrical modulation of the inter-subunit interaction energies, emphasizing the importance of dynamics-based structural profiling in understanding the asymmetrically oriented chaperonin function [8]. Although extensive structural studies, including X-ray crystallography and single-particle cryo-EM, have provided insights into the atomical detail views of the CCT complex, real-time dynamics analyses of type II chaperonins are still yet to be well-documented [10,15,16,17,18,19,20].

Diffracted X-ray tracking (DXT) is a powerful single-molecule technique that investigates functional protein dynamics with nanoscale precision, utilizing nanocrystals and synchrotron X-rays [21,22,23,24] (Figure 1E). By analyzing nanocrystal’s diffraction spot trajectories, the DXT analysis detects atomic-scale protein motion with microsecond to millisecond time resolution, making it ideal for studying cooperative motion in multimeric proteins (Figure 1F). Through DXT analysis, research on a type II chaperonin from hyper-thermophilic archaea revealed rapid and coordinated conformational changes within 2 s of ATP binding, shedding light on the functional folding process [25,26]. However, the molecular dynamics of mammalian chaperonins are largely unknown.

In this study, we employed DXT technique to track the single-molecular dynamics and investigate the microsecond-order kinetic analysis of the human CCT complex.

## 2. Results

### 2.1. DXT Analysis of the CCT Complex

To perform single-molecule dynamics analysis using DXT, the CCT complex needed to be immobilized on the substrate and nanoparticles (Figure 1E). Given that the CCT1 subunit adopts a structurally diagonal position in the CCT steric complex, inserting the tag into the CCT1 subunit would place it on the diagonal, rendering it most suitable for DXT analysis of the CCT complex (Figure 1B,G). For this purpose, the PA-tag, capable of insertion into loop structures, was utilized in tagging three different surface-exposed loop regions of the CCT1 subunit (Figure 1C): the loop region of the E-domain (No. 1), the loop region between the E-domain and I-domain (No. 2), or the loop region of the A-domain (No. 3), to detect differences in motion in structurally distinct regions [9] (Figure 1D). The location of No. 2 is relatively close to the ATP binding region and is connected to the nucleotide sensing loop (NSL) via an α-helix [27] (Figure 1D). We expressed PA-tagged CCT complexes in HEK293T cells and purified them. The structure of these purified complexes was then verified using electron microscopy, as shown in Appendix A. To confirm the antibody-mediated immobilization of the CCT complex on the surface, we employed Atomic Force Microscopy (AFM) (Figure 1H). Each round-shaped particle, which represents an individual CCT complex, was properly immobilized on the antibody-coated surface.

DXT employs broadband X-ray irradiation based on Bragg’s law. The diffraction rings from the X-ray beams, known as Debye–Scherrer rings, broaden according to their energy bandwidths. The motion of diffraction spots is analyzed in two rotational axis views, including tilting (θ) and twisting (χ) angles (Figure 1F). In this experiment the DXT measurements were conducted at the NW14A beamline from PF-AR at KEK (Tsukuba, Japan). Data were recorded at 100 milliseconds per frame, and the total measurement time was 50 s. The number of analyzed trajectory points in this study were summarized in Table 1. 

### 2.2. Dynamic of the CCT Complex in Different Structural Regions and Conditions

We analyzed the overall structural dynamics of three differently tagged CCT complexes (No. 1~No. 3) under three distinct conditions: in the absence of ATP (referred to as “Control”), in the presence of 1 mM ATP (referred to as “ATP”), and in the presence of 1 mM ATP with substrate (1 μM denatured Actin) (referred to as “ATP+dActin”). Firstly, we examined the time-averaged mean square displacement (MSD) curves using the obtained diffraction tracks (Figure 2A). Upon analyzing the resulting diffraction spots in two rotational axes, namely tilt (θ) and twist (χ) angles, we observed that the MSD values were larger for twist (χ) angles compared with tilt (θ) angles in all three regions (No. 1~No. 3) and under all three different conditions (Figure 2B). This suggests that the CCT complex exhibits a greater tendency for twisting or rotational movement rather than tilting motion.

Therefore, we focused on the twist (χ) direction and examined variations in kinetics across distinct structural regions. Under the Control condition, No. 1(χ) and No. 3(χ) showed more enhanced MSD values than No. 2(χ), and this difference disappeared in the presence of ATP, while in the condition of ATP+dActin, No. 2(χ) had enhanced MSD values compared with No. 1(χ) and No. 3(χ) (Figure 2B). The motions at No. 1(χ) and No. 3(χ) were suppressed by ATP and ATP+dActin; however, the motion at No. 2(χ) was rather enhanced by them (Figure 2C). 

### 2.3. The Probability Distributions and the Mean Transitions of the Dynamics

Next, we employed a Gaussian plot of detected trajectories to visualize the individual spots’ movements at each time point and to investigate the properties of each motion under three conditions [28]. The probability distributions of twist (χ) were plotted as a function of time intervals (Δt = 0.1, 5, 10 s) (Figure 3A–C). The displacement distributions of all conditions were fitted with single Gaussian curves, suggesting stochastically uniform and non-biased conformational dynamics under diverse conditions. As the time intervals (Δt) increased from 0.1 to 5 and 10 s, the standard deviations (σ) of the Gaussian function also increased from a value in the second half of 0.5 to a value in the first half of 0.6, except for the Control condition in No. 2.

The mean (μ) values of the Gaussian function plotted over the time interval (Δt), referred to as the “mean transition,” represent the rotational bias generated in the CCT complex dynamics (Figure 3D). The mean transitions of the three different regions exhibited distinct patterns under the Control condition. At a time interval (Δt) of 0.1 s, the mean (μ) values in all three regions were approximately −0.3 mrad, indicating a negative rotational bias within the 0.1 s analysis interval for the CCT complex. As the time interval (Δt) elapsed, this bias gradually diminished, and after approximately 3 and 5 s, No. 3 and No. 1 transitioned to positive values, respectively, while No. 2 exhibited negative values during the 10 s time interval.

The mean transitions under the ATP and ATP+dActin conditions did not differ significantly from those under the Control condition in the three regions. At a time interval (Δt) of 0.1 s, the mean (μ) values in all three regions were around −0.25 to −0.3 mrad. Over time, this bias also gradually diminished, and after approximately 4~5 s, the mean (μ) values transitioned to positive values, and the biases were enhanced in the positive direction.

### 2.4. Lifetime Classification of the Trajectories

To further understand and classify the behavior of the trajectories, we introduced the concept of “lifetime”, referred to here as a “lifetime classification” [29]. Lifetime is defined as the period from the emergence to the vanishing of a diffraction spot within the observation area, recorded using synchrotron radiation within a specific energy range. Fast-moving proteins display shorter durations for their diffractions, whereas slow-moving proteins exhibit longer durations. Thus, the duration of the diffractions effectively represents the overall movement (lifetime: LT) of the respective target protein.

MSD curves were calculated for each LT data, revealing that the MSD values were consistently larger for twist (χ) angles compared with tilt (θ) angles in all three regions (No. 1~No. 3) across all conditions, consistent with the data of Figure 2B (Appendix A). Here, we focused on the movement in the twist (χ) direction and observed kinetic changes in the angles of the MSD curves, i.e., angular diffusion coefficients (D), at the time interval (Δt) of 0.1 s (Figure 4B). The D values demonstrated a general trend where the longer the lifetime, the less the angles expanded, aligning with the concept of the lifetime classification (Appendix A). Comparing the data of each tag region, under the Control condition, the D values were enhanced in No. 1(χ) and No. 3(χ) than in No. 2(χ) among the four LT groups. In ATP and ATP+dActin conditions, differences among three regions were especially observed in the LT1 group: in ATP conditions, No. 2(χ) and No. 3(χ) showed enhanced D values than No. 1; in ATP+dActin conditions, No. 1 showed enhanced D values than No. 2(χ) and No. 3(χ). 

A detailed analysis of the mean square displacement (MSD) curves for the LT1 group in each tag region revealed that the MSD curves for No. 1(χ), No. 2(χ), and No. 3(χ) exhibited the least expansion under ATP, Control, and ATP+dActin conditions, respectively (Figure 4C). The MSD curves for No. 1(χ) and No. 3(χ) showed enhanced expansion under both Control and ATP+dActin conditions and both Control and ATP conditions, respectively, with no significant differences.

The mean transitions showed that those of the LT1 group were consistently positive under all conditions (Figure 4D). As the lifetime increased, the negative values became more pronounced in the initial phases (Δt = 0.1~0.2 s). This suggests that the rotational bias observed in Figure 3D is mainly derived from the longer lifetime groups. 

## 3. Discussions

In this study, we conducted real-time molecular dynamics mapping to investigate the dynamics of three surface-exposed loop regions within the CCT1 subunit of mammalian type II chaperonins at the millisecond level. To our knowledge, this is the first study to explore this aspect.

According to the primary mechanism, the chaperonins capture an unfolded protein through the apical (A) domain in an open state (Figure 5, State I). ATP binding causes the apical domains to rotate, for example, about 45% in the case of the archaeal chaperonin Mm-Cpn, resulting in a more symmetric and compact open conformation [15,30,31] (Figure 5, State II).

Unfolded actin has been observed to interact with the CCT complex in a variety of ways, with its unstructured segment extending into the central chamber [4,17,32,33] (Figure 5, State II). Some studies have suggested that actin can bind to the CCT1 subunit, either near the E domain or around the connecting regions between the I and A domains [32,33]. Another study found that denatured actin had a primary association with CCT4/2/5/7 subunits [4,17]. These variations in binding modes may suggest differences in the folding states of unfolded actin, including differences in surface electrostatic polarities and hydrophobic properties.

**Figure 5 ijms-24-14850-f005:**
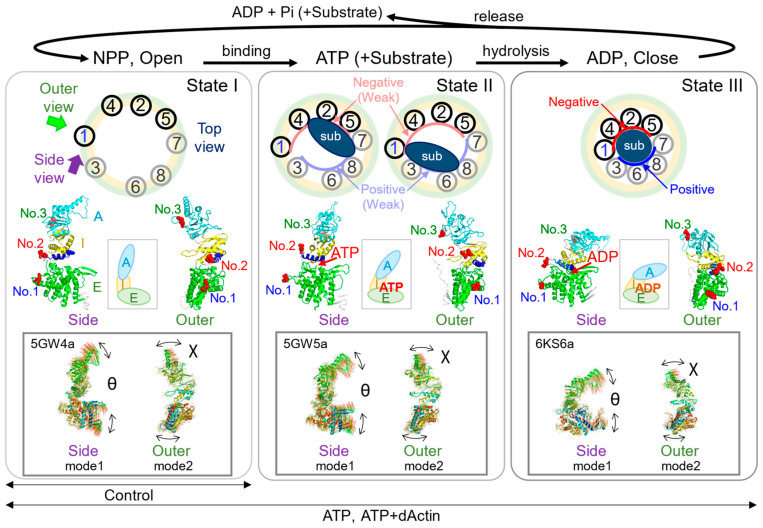
Dynamics of the CCT complex during the functional cycle. The figure summarizes the structural changes in the CCT functional cycle. State I (**left**): The chaperonin is in an open conformational state, either nucleotide-free or nucleotide-partially preloaded (NPP). State II (**middle**): ATP and/or the substrate are bound, resulting in a slightly more compact structure than State I. State III (**right**): ATP is hydrolyzed to ADP, leading to the chaperonin adopting a closed state. Following the release of ADP and Pi or the substrate, the CCT structure transitions from a closed to an open state, returning to State I. The schematic models at the top show a top view of each state. The outer (light green) and inner (light yellow) circles illustrate the structure transition to a more compact structure as the chaperonin shifts from an open to a closed state. The binding modes of substrate, i.e., unfolded or partially folded actin, are indicated by dark blue ovals or circle in States II and III, respectively. The “sub” above dark blue ovals or circles stands for substrate. In State III, the CCT1/4/2/5 cluster and the CCT7/8/6/3 cluster create regions with high ATP affinity and negative charge (red inner curve) and low ATP affinity and positive charge (blue inner curve), respectively. Light red and blue curves in State II indicate the relatively low intensity electrostatic polarities. To the right and left of the middle of each state, PDB structural models are presented, offering views from the outer (indicated by the green arrow in State I) and the side (indicated by the purple arrow in State I). The protein structure model in State I is based on PDB ID 5GW4 [15], whereas structures in States II and III are derived from previously published work [8]. The middle schematic models of the middle row illustrate the simplified changes as the subunit transitions to State I, State II, and State III. The squares shown at the bottom row of each state show the displacement vectors of atoms of individual normal modes for which data have been published as ProMode-Elastic database, using mode 1 (side view) and mode 2 (outer view), together with the directions of axes of χ and θ in each state [34,35,36,37]. PDB codes cited are shown in the upper left corner of each square.

ATP hydrolysis triggers the closure of the lid, releasing the bound substrate into the central chamber. The central chamber has distinct electrostatic polarities in the closed state, which would facilitate the folding process [8,38,39] (Figure 5, State III). The release of phosphate and ADP causes the chamber to reopen, releasing the substrate in its refolded form [40]. 

Previous studies have estimated that the CCT complex consumes one ATP molecule every 10–14 s [41,42]. This rate can be accelerated in the presence of unfolded substrate [43]. It is not fully understood how ATP molecules are hydrolyzed individually or concertedly among the eight CCT subunits during the CCT functional cycle, but the analyses performed with a time interval of 10 s in Figure 2 and Figure 3 would fit within the timeframe of one cycle. 

Based on the above mechanism, we hypothesized that the CCT complex takes on a relatively larger and more relaxed conformation when nucleotide is not present (State I), compared with the state when nucleotide is bound (States II or III). Consistent with this hypothesis, our mean square displacement (MSD) analysis of No. 1 and No. 3 revealed expanded movements under the Control condition compared with the ATP and ATP+dActin conditions (Figure 2C). The linearity of the MSD curves was almost linear up to about 8 s under all conditions, suggesting that the A-domain and E-domain experience unrestrained or flexible movement under the Control condition. Nevertheless, no notable differences were noted in these regions under the ATP and ATP+dActin conditions.

No. 2, which corresponds to the connecting loop between the E-domain and I/A-domains, exhibited distinctive motion patterns with progressively expanding mean square displacement (MSD) curves across the Control, ATP, and ATP+dActin conditions (Figure 2C). While it might have been expected that No. 2 would display a similar degree of MSD curve expansion as observed in No. 1 and No. 3 under the Control condition, its movement was notably constrained compared with No. 1 and No. 3. This limitation can be due to its location at the pivotal junction linking the E-domain and the I/A-domains (Figure 1D).

Normal mode analysis (NMA) based on elastic-network model (ENM) using dihedral angles can capture the rotational motions of the protein backbone and side chains, which are often biologically relevant for large biomolecular complexes [37,44,45,46]. The results of NMA from multiple studies support our finding that the region around No. 2 could function as an axis of rotation for the subunit [34,45,47] (Figure 5).

No. 2 is also situated near the nucleotide sensing loop (NSL), which houses a lysine residue (K159 within the CCT1 subunit) (Figure 1D). This residue plays a central role in monitoring the rate of ATP hydrolysis and the nucleotide state [27,38,48]. Nucleotides can wedge in the ATP-binding region, weakening the region of No. 2 as the axis of rotation and making the CCT complex more compact. This can be reflected in the less expanded MSD curves of No. 1 and No. 3 in the presence of nucleotides. On the other hand, the MSD curves of No. 2 were more enlarged compared with the Control condition under the ATP and ATP+dActin conditions (Figure 2C). Hence, the dynamics of No. 2 can be affected by the ATP binding and hydrolysis that occurs in the NSL.

The Gaussian distribution plot of the tracked trajectories suggests that the conformational dynamics of the system were stochastically uniform and relatively unbiased (Figure 3A–C). The standard deviations (σ) of the Gaussian distribution were generally increased with longer time intervals (Δt), which reflects the broadening of the distribution of the twist angle (χ). A notable exception to this trend was observed in the Control condition for No. 2, which is consistent with the data of lesser MSD expansion shown in Figure 2B. The same uniqueness of No. 2 was observed not only in the mean transition, which represents the rotational bias of the CCT complex (Figure 3D), but also in the lifetime filtering technique (Figure 4). Under the Control condition, No. 2(χ) had lower MSD values than No. 1(χ) and No. 3(χ) in all LT groups (Figure 4A and Appendix A). Furthermore, in the presence of nucleotides, the MSD curves of No. 2(χ) were expanded in all LT groups.

In summary, in the absence of nucleotides, the region around No. 2 may function as an axis of rotation for the E-domain (No. 1) and I/A-domains (No. 3), as shown in Figure 5 (State I). This allows No. 1(χ) and No. 3(χ) to have a wider range of motion, as seen by their expanded MSD curves under control conditions (Figure 2B and Figure 4A). However, in the presence of nucleotides, the nucleotides may act as a wedge in the ATP-binding region, weakening the role of the region around No. 2 as the axis of rotation and causing the CCT complex to adopt a more compact structure (Figure 5, States II and III). This is reflected in the less expanded MSD curves of No. 1 and No. 3 under these conditions (Figure 2C and Figure 4C). This compact structure may help to stabilize the subunit in its substrate binding and functional conformations. Meanwhile, the dynamics of No. 2 may be expanded by the influence of the binding and hydrolysis of nucleotides. Only after the release of the wedges, i.e., nucleotides (ADP), the CCT complex can again adopt an open structure (Figure 5, State I). 

Given the double-ring structure of the CCT complex, the movement of a nanocrystal in DXT analysis was intertwined with the conformational changes in both rings. Although the twisting motions of the rings occur in opposite directions, their twisting can be observed in the same rotational direction of the nanocrystal due to the symmetrical presence of each tag position in the 3D structure (Figure 1F–G). Currently, it is unfeasible to dissect these motions within the present experimental setup [25]. Prior structural investigations suggest the existence of weaker inter-ring subunit interactions because of the asymmetry between the inter-ring configurations [49,50,51,52]. It is plausible that co-chaperones, such as phosducin-like proteins stored within the chamber of the CCT complex along with unfolded actin, may regulate inter-ring communication [32,33]. Since no co-chaperones were introduced in this study, the observed dynamics would be primarily attributed to the ring on one side. 

Further studies are needed to clarify the dynamic changes in each position that were not fully clarified in this study. For example, the mean transition results consistently indicated a movement towards the minus direction by approximately 0.3 mrad at a time interval (Δt) of 0.1 s (Figure 3D). While this rotational bias can be derived from the longer lifetime groups (Figure 4D), the rapid structural dynamics of the CCT complex are not entirely captured within a 0.1 s interval. This emphasizes the need for further analysis to track even faster movement dynamics, as well as the effect of the substrates on the dynamics of the CCT complex. 

## 4. Materials and Methods

### 4.1. DNA Construction and Transfection

A full-length human CCT1 gene was subcloned into pDONR221, and a PA-tag sequence (GVAMPGAEDDVV) was introduced at three different loop positions using the inverse PCR system (Toyobo Biotech, Osaka, Japan). The positions of the PA-tags on the CCT1 protein were between 142E and 143L (No. 2), 341E and 342A (No. 3), or 477N and 478P (No. 1). Subsequently, these three open reading frames were transferred to the pMXs-gw-IP expression vector using the Gateway LR reaction system (Thermo Fisher Scientific, Waltham, MA, USA), following the manufacturer’s protocol.

HEK 293T cells were cultured in a tissue culture incubator maintained at 37 °C with 5% CO_2_ and over 90% humidity, using DMEM (high glucose) supplemented with 10% fetal calf serum (GIBCO, Waltham, MA, USA). Cells were transfected with the constructs using Lipofectamine 2000 reagent (Thermo Fisher Scientific) and then subjected to selection with 1 μg/mL puromycin. Drug-resistant cells from each transfected population were subsequently pooled to establish polyclonal cell lines expressing the PA-tagged CCT1 variants. Immunoblotting was performed on each cell line using the anti-PA tag rat monoclonal antibody (clone NZ-1, FUJIFILM-Wako, Osaka, Japan) to confirm the expression of the desired PA-tagged CCT1 variants.

### 4.2. Purification of CCT Complex

Approximately 5 g of cell pellets were homogenized in 50 mL of homogenization buffer, consisting of 5 mM MgCl_2_, 0.1 mM EDTA, 20 mM HEPES-KOH (pH 7.4), 30 mM NaCl, 10% glycerol, 1 mM DTT, 0.02% Tween 20, and 0.5 mM AEBSF. The homogenate was clarified by centrifugation at 14,900 g for 20 min. ATP was added to a final concentration of 1 mM, and then (NH_4_)_2_SO_4_ was added to 15% saturation. Precipitated proteins were discarded by centrifugation at 14,900 g for 30 min. Next, (NH_4_)_2_SO_4_ was added to 50% saturation to the supernatant, and precipitated proteins were pelleted by centrifugation at 14,900 g for 30 min. The resulting pellet was carefully dissolved in Buffer A, containing 5 mM MgCl_2_, 0.1 mM EDTA, 20 mM HEPES-KOH (pH 7.4), and 30 mM NaCl with 10% glycerol, and then dialyzed three times (1 h each time) against Buffer A at 4 °C. The dialyzed extract was centrifuged for 1 h at 14,900× *g* to remove any precipitated material. Subsequently, 1 mM ATP was added to the supernatant, and the solution was incubated for 20 min before being loaded onto 4 × 1 mL pre-equilibrated HiTrap heparin columns (Cytiva, Marlborough, MA, USA). The HiTrap Heparin columns were run using Buffer A and Buffer B, where Buffer B contained Buffer A plus 1 M NaCl. After loading the samples, the columns were washed with 30 mL of Buffer A at a flow rate of 0.5 mL/min. A linear gradient from 0 to 60% B at a flow rate of 0.4 mL/min was conducted, and all fractions were run on an SDS-PAGE. CCT-containing fractions were pooled and proceeded to subsequent gel filtration (HiLoad 16/60 Superdex 200 pg column) (Cytiva) equilibrated with Buffer A. Purified samples were concentrated, stored at 4 °C, and used within 3 or 4 days.

### 4.3. DXT Measurements

A 50 μm-thick polyimide film (Kapton, Du Pont-Toray, Tokyo, Japan) served as the substrate surface for DXT and was coated with chromium (10 nm) and gold (25 nm) via vapor deposition. To immobilize the PA-tagged chaperonins, the gold-coated film surface was treated with anti-PA tag antibodies (0.1 mg/mL) in PBS (pH 8.0) overnight at 4 °C. Afterward, the resulting antibody-modified substrate was thoroughly washed with PBS and Buffer A. Subsequently, PA-tagged chaperonin solutions (0.2 mg/mL) in Buffer A were applied to the modified substrate for 1 h at 4 °C. The chaperonin-coated surfaces were rinsed with the same buffer and then reacted with ZnO nanocrystals (NP-ZNO-3, EMJAPAN, Tokyo, Japan) modified with anti-PA tag antibodies through thiol-ZnO chemistry. Unbound ZnO conjugates were washed away twice with Buffer A.

The experimental setup involved encapsulating the samples within a 20 μL chamber using 5 μm-thick polyimide films, which were sandwiched by stainless steel frames and screw-clamped. Specific experimental buffers, including Buffer A (Control), Buffer A supplemented with 1 mM ATP (ATP), and Buffer A supplemented with 1 mM ATP and 1 μM denatured actin (rabbit skeletal muscle; Cytoskeleton, Inc., Denver, CO, USA) (ATP+dActin), were used for the experiments. The actin protein was denatured in a 4M guanidine hydrochloride solution at a concentration of 200 μM and was then diluted 200-fold (final concentration 1 μM) just before the experiment. 

The dynamics of a single protein were observed by tracking the trajectories of Laue spots emitted by ZnO nanocrystals, following the methodology described previously [53]. In brief, the NW14A beamline (Photon Factory Advanced Ring, PF-AR at KEK, Tsukuba, Japan) was utilized to generate high-coherence bright white synchrotron X-rays, with a beam size of 100 μm × 250 μm, to monitor Laue diffraction spots from ZnO nanocrystals on the chaperonins. The data were recorded at 12.5 ms/frame intervals, and the diffracted photons were counted using a 2D photon-counting detector, PILATUS 100 K array (DECTRIS, Baden-Dättwil, Switzerland; 0.172 mm per pixel). Data acquisition was performed at 100 ms/frame intervals, with 5000 frames per spot being measured. For each sample, diffractions of ZnO (110), (103) at 2 spots were recorded, spaced 0.2 mm apart. All DXT measurements were conducted at room temperature.

### 4.4. Data Analysis for DXT

Each diffraction spot was tracked by TrackPy (v0.6.1, https://zenodo.org/record/7670439, accessed on 25 May 2023) after correcting the background. Trajectories were analyzed using a custom software written by python. For each spot trajectory, the initially appeared time and position (in χ and θ angles) were defined as the starting time and position, respectively. Next, the displacement of angles (χ and θ) from the starting position was plotted as a function of the elapsed time from the starting time (time lag). The ensemble-averaged squared displacements as a function of time lag were then used to produce MSD curves [29].

To analyze the lifetime classification, trajectories were divided into four different groups based on their durations: immediate (lifetime (LT) < 1.5 s: LT1), short (1.5 s ≤ LT < 4.0 s: LT2), medium (4.0 s ≤ LT < 7.5 s: LT3), and long (7.5 s ≤ LT: LT4). These trajectory groups were extracted from the 100 ms/frame recording data and analyzed separately. The diffractions originating from fast-moving proteins exhibited a brief duration between their appearance and disappearance in the recording area, whereas those from slow-moving proteins had a longer duration. 

At the time interval of 0.1 s, MSD curves were fitted with least-squares fitting to the following equation, MSD = 4Dt + A, where MSD is the mean square angular displacement, D is the angular diffusion constant, t is time interval, A is intercept of the MSD curve.

### 4.5. AFM Analysis

The AFM technique was used to visualize the immobilization of the CCT complex on the surface coated with anti-PA tag antibody, which was covalently immobilized on mica, following the procedure described in the literature [54,55]. Briefly, 2 μL of 0.01% APTES (Sigma Aldrich, Burlington, MA, USA) diluted in ultra-pure water was applied to mica and incubated for 3 min, followed by two rounds of washing with ultra-pure water to remove any unbound APTES. Next, 0.2% glutaraldehyde was added to the APTES-modified mica and incubated for 5 min. After washing away the unbound glutaraldehyde with PBS, the anti-PA tag antibody (FUJIFILM Wako Chemicals, Osaka, Japan) was applied. Any unbound antibodies were removed, and free aldehyde groups were neutralized by washing with tris-buffered saline. Subsequently, the CCT complex was incubated in Buffer A for 10 min before proceeding to AFM observation. The custom-made high-speed AFM, equipped with the electro-beam deposition probe on the top of the ultra-short cantilever (BL-AC10DS-A2, Olympus, Tokyo, Japan), was operated following the procedure described in the literature [56,57,58].

### 4.6. Electron Microscopy

The CCT complex sample was applied onto thin carbon films supported by copper mesh grids, which were rendered hydrophilic in advance by glow discharge under low air pressure. The samples were negatively stained with EM Stainer (Nisshin EM, Tokyo, Japan), blotted, and air-dried. They were observed using a JEM-1230 transmission electron microscope (JEOL, Tokyo, Japan) at an acceleration voltage of 100 kV. The images were recorded using a TVIPS F114T CCD camera (TVIPS, Oslo, Norway).

## Figures and Tables

**Figure 1 ijms-24-14850-f001:**
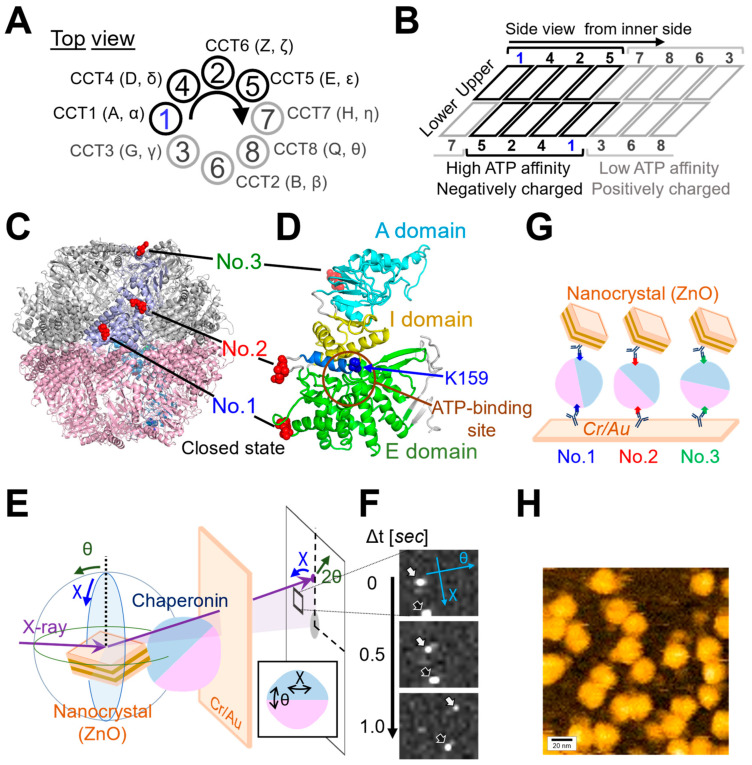
Organization of the CCT Complex and Diffracted X-ray Tracking (DXT) measurement. (**A**) Top view and (**B**) side view from the inner side, illustrating the schematic representation of the CCT Complex. CCT4/2/5 subunits exhibit high ATP affinity and carry a negative charge, whereas CCT8/6/3 subunits have low ATP affinity and a positive charge. (**C**) Ribbon diagram of the closed state of the CCT complex. The light purple color indicates the region of the CCT1 subunit. The red space-fill models indicate the three insertion sites of the PA-tag employed in this study. The structure is based on the simulated human CCT complex, as previously detailed in our prior publication [8]. (**D**) Ribbon diagram of the CCT1 subunit demonstrating the common domain structure shared by all eight subunits. Equatorial domain (E domain) is represented in green, intermediate domain (I domain) in yellow, and apical domain (A domain) in cyan. The region encompassing the ATP binding site is delineated by brown circles. Lysine 159 (K159) is presented in a blue surface view. Positioned at the ATP binding site, this evolutionally conserved lysine residue is known for its crucial role in monitoring ATP hydrolysis rate and nucleotide status. A connecting α-helix in blue links No. 2 and K159. (**E**) Diffraction spots produced by ZnO nanocrystals attached to the Chaperonin molecules were tracked using high-coherence bright white synchrotron X-rays generated by the NW14A beamline at PF-AR, KEK. The angular displacement changes in the chaperonins were tracked along the two-dimensional axes of twist and tilt (χ and θ). The inset diagram schematically shows the directions of axes of χ and θ. The time-resolved diffraction images were analyzed separately, based on the χ-θ coordinates. (**F**) Representative time-resolved trajectories, indicated by arrows, are shown. (**G**) The anti-PA tag antibodies [9] were immobilized on a gold- and chromium-coated Kapton surface via thiolate bonds formed by the cysteine residues on the antibodies. On top of the anti-PA tag antibody coated surface, each CCT complex was placed, and the other side was attached with anti-PA tag antibody-labeled nanocrystal (ZnO). For simplicity, this figure shows only the antibodies that interact with the CCT complex. (**H**) Atomic Force Microscopy (AFM) confirmed immobilization of CCT molecules on the anti-PA tag antibody coated substrate, with No. 2 used as a representative example. AFM operates a tiny probe, on a flexible cantilever beam, to interact with a sample and adjust its height based on the cantilever deflection. By scanning the probe across various points of the sample’s surface, a detailed nano-scale image of the sample can be generated. Brighter areas indicate higher elevations. Scanning area, 200 nm × 200 nm (100 pixel × 100 pixel). Scanning rate, 0.5 second (s) per frame. Z-range, −3 to 15 nm (0 nm at mica surface). The bar represents 20 nm.

**Figure 2 ijms-24-14850-f002:**
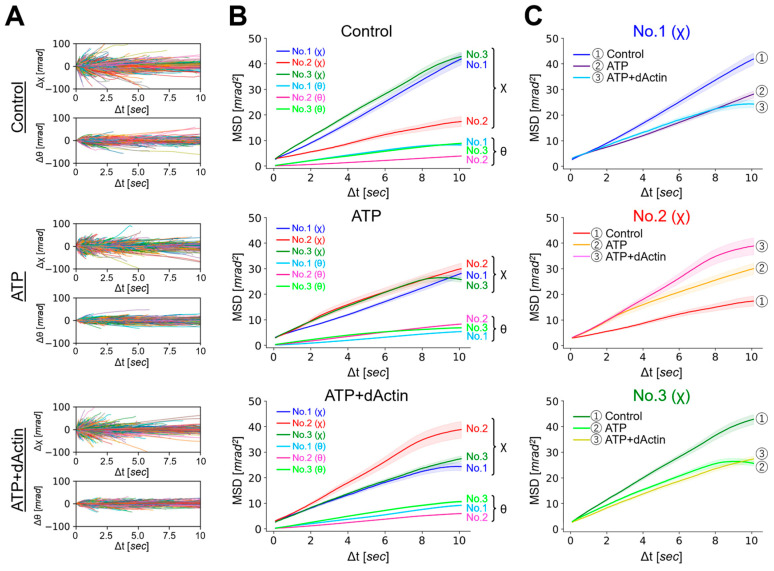
Dynamics of the CCT Complex in Different Structural Regions and Conditions. (**A**) Rotational angular displacement along the χ (upper) and θ (lower) axes is visualized over a 10 s duration. Traces from samples of No. 1 under the Control, ATP, and ATP+dActin conditions are represented in the top, middle, and bottom panels, respectively. (**B**) Mean square displacement (MSD) curves illustrate the motion of the CCT complex along the θ and χ axes under the Control, ATP, and ATP+dActin conditions. The No. 1, No. 2, and No. 3 samples are color-coded in blue, red, and green, respectively. The 95% confidence levels are indicated with shading. (**C**) Focusing on the twist (χ) direction, MSD curves for each region under the Control, ATP, and ATP+dActin conditions are shown.

**Figure 3 ijms-24-14850-f003:**
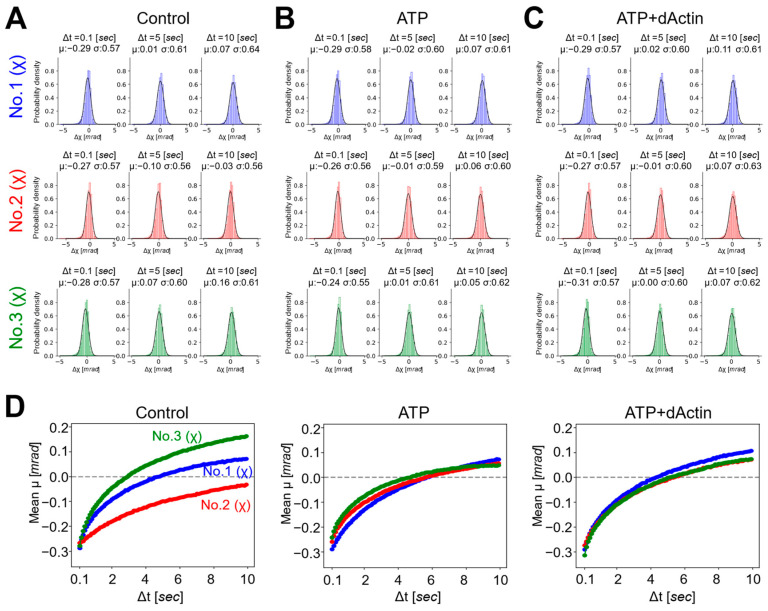
Probability distributions and mean transitions of the CCT complex. (**A**–**C**) Probability distributions of the twist (χ) angle for No. 1, No. 2, and No. 3 were plotted as a function of time intervals (Δt = 0.1, 5, 10 [sec]) under the (**A**) Control, (**B**) ATP, and (**C**) ATP+dActin conditions. (**D**) Mean transitions for the twist angle (χ) under the conditions of Control (left), ATP (middle), and ATP+dAction (right) were plotted over a 10 s duration.

**Figure 4 ijms-24-14850-f004:**
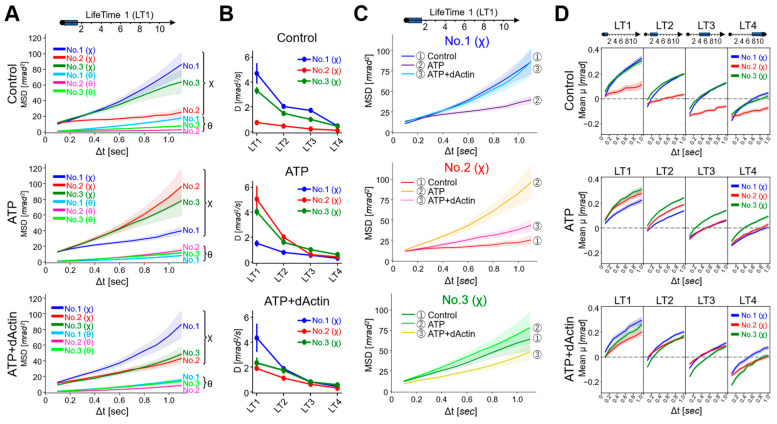
Dynamics for Each Lifetime Group. (**A**) MSD curves of the LifeTime 1 (LT1) group under the Control (upper), ATP (middle), and ATP+dActin (lower) conditions are shown. The 95% confidence levels are indicated with shading. (**B**) The slope values (Angular diffusion coefficient: D (mrad^2^/s)) of the MSD curves at the time interval (Δt) of 0.1 s under three conditions for the four LT groups were calculated using the following equation: MSD = 4Dt + A, where MSD is the mean square displacement, D is the angular diffusion constant, t is the time interval, and A is the intercept of the fitting line. The slope values and standard deviation (S.D.) data are summarized in Appendix A. (**C**) The MSD curves of three regions in the LT1 group are shown under the Control, ATP, and ATP+dActin conditions. The 95% confidence levels are indicated with shading. (**D**) The mean transitions based on lifetime classification for the twist angle (χ) are shown under the conditions of the Control (upper), ATP (middle), and ATP+dActin (lower) over a 1 s duration.

**Table 1 ijms-24-14850-t001:** Summary of the number of trajectories analyzed in this study. The respective percentages for the individual conditions are shown in parentheses.

Condition		Total	LT1	LT2	LT3	LT4
Control	No.1	25,808	11,486 (83.4)	1334 (9.7)	438 (3.2)	515 (3.7)
	No.2	13,773	18,334 (82.4)	2470 (11.1)	738 (3.3)	701 (3.2)
	No.3	22,243	25,515 (80.7)	3570 (11.3)	1211 (3.8)	1310 (4.1)
ATP	No.1	31,606	19,342 (82.8)	2497 (10.7)	710 (3.0)	807 (3.5)
	No.2	23,356	23,839 (83.2)	3044 (10.6)	891 (3.1)	893 (3.1)
	No.3	28,667	13,634 (84.0)	1671 (10.3)	473 (2.9)	447 (2.8)
ATP+dActin	No.1	16,225	15,907 (81.5)	2156 (11.0)	761 (3.9)	704 (3.6)
	No.2	19,528	13,731 (81.9)	1897 (11.3)	590 (3.5)	555 (3.3)
	No.3	16,773	11,486 (83.4)	1334 (9.7)	438 (3.2)	515 (3.7)

## Data Availability

The data that support the findings of this study are available from the corresponding author upon reasonable request.

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
