# Peer review of "Molecular Dynamics Mappings of the CCT/TRiC Complex-Mediated Protein Folding Cycle Using Diffracted X-ray Tracking"

_ijms, 2023, doi:10.3390/ijms241914850_

Round 1

Reviewer 1 Report

This manuscript is well described. I would like the authors to explain the binding effects of ZnO nanocrystals. Where (what the residues) do they react with? How much of the ratio (stoichiometry)?

Minor;

In the section 4.2, the numbers of MgCl2 and (NH4)2SO4 should be described as subscript.

Author Response

# Comments from Reviewer1

Comment: I would like the authors to explain the binding effects of ZnO nanocrystals. Where (what the residues) do they react with? How much of the ratio (stoichiometry)?

Reply:

ZnO nanocrystals as well as Au (Au deposition films in this study) are known to interact strongly with thiol groups. This interaction is chemically known as the "thiolate bond”. The chemical names for the bond between zinc and sulfur and the bond between gold and sulfur are zinc sulfide (ZnS) and aurous sulfide (AuS), respectively.

In this study, the Au deposition films and ZnO nanocrystals were pre-interacted with the anti-PA tag antibodies. This allows the thiol groups of cysteine residues on the antibody surface to interact with the Au deposition films and ZnO nanocrystals. The Au deposition films and ZnO nanocrystals then interact with Chaperonin (CCT complex) via the antibodies. Since the CCT complex has two PA tags on the diagonal of the structure as shown in Figure 1G, one PA tag on one side of the CCT complex is bound to the Au deposition films, while the ZnO nanocrystal is bound to the PA tag on the other side of the CCT complex. One point missing from Figure 1G is that the Au film and ZnO nanocrystal do not show the binding of many antibodies on the surface of the anti-PA tag antibody.

Thus, the legend of Figure 1G was modified as follows.

(G) The anti-PA tag antibodies were immobilized on a gold- and chromium-coated Kapton surface via thiolate bonds formed by the cysteine residues on the antibodies [9]. On top of the anti-PA tag antibody coated surface, each CCT complex was placed, and the other side was attached with anti-PA tag antibody-labeled nanocrystal (ZnO). For simplicity, this figure shows only the antibodies that interact with the CCT complex.

It is also possible that the interaction of ZnO nanocrystals with chaperonins has some effect on the kinetic state of the chaperonins. Therefore, in this study, we do not discuss the absolute kinetic parameters, but focus on the relative kinetic changes by comparing the results of kinetic analyses in which tags were inserted at three different positions. As noted in the text, the observed distribution of motion obtained in the results of this study showed a typical Gaussian distribution shape, which suggests that no compound motion was observed and that it is highly likely that the single ZnO nanocrystals observed motion that was attached to a single chaperonin molecule.

Minor;

Comment: In the section 4.2, the numbers of MgCl2 and (NH4)2SO4 should be described as subscript.

Reply:

Thank you for pointing this out. The numbers were corrected. I have highlighted the corrected text in red in the manuscript.

We appreciate your time and effort in providing us with valuable feedback.

Reviewer 2 Report

Kazutaka Araki et al. demonstrated the effect of ATP and substrate binding on CCT subunit motions using diffracted X-ray tracing, validated by AFM, and proposed that the ATP could wedge into the intermediate domain of the CCT subunit, limiting its mobility and stabilizing the closed conformation. I believe this study provided significant experimental insights into the mechanism of action of the allosteric cycle of CCT and should be published after minor revision. I have the following suggestions to improve the manuscript:

1. In Figure 1, the authors could add an additional diagram showing the chi and theta angles on a single CCT subunit with twisting and nodding motions to provide more physical intuition.

2. For Figure 1H, it would be helpful to provide some more descriptions of what the image shows and how it confirms the DXT result for people who are unfamiliar with AFM experiments.

2. In the discussion and Figure 5, the authors hypothesized that the limited mobility of No.2 can be explained by the role of the intermediate domain (tagged with No.2) as the rotation axis for CCT subunits using normal mode analysis. Another computational study of TRiC/CCT using elastic network analysis of Cryo-EM structures, Yan Zhang et al. Progress in Biophysics and Molecular Biology (2021) 160: 104-120, also showed the role of the intermediate domain as the rotation axis in both the twisting and nodding motion (supplementary movies), which might be relevant for this study.

3. Several other elastic network model (ENM) studies on chaperones and large biomolecular complexes could enrich your discussion. I recommend considering the following publications:

(a) Ivet Bahar et al., Biophysical Journal (2015) 109(6)

(b) Pablon Chacon et al., J. Mol. Bio. (2003), 326

(c) She Zhang et al., Bioinformatics (2021), 37(20)

4. Figure 5 as the centerpiece of this study is very informative but could be somewhat overwhelming for some readers. I have the following suggestions to make it more accessible:

(a) Label the substrate circle/oval with “substrate” or “actin” instead of describing it in the caption.

(b) Label the PA tags as No.1, No.2, and No.3, consistent with Figure 1 and the texts.

(c) Perhaps label the light blue (red) as weak positive (negative) in the figure or use legends. Otherwise, it is easy to miss this among the long captions.

(d) For the open state, although normal modes are bidirectional, it may be clearer for readers to either change the arrows of the protein structure to show an inward movement, consistent with the other two panels, or point this out in the caption.

Author Response

# Comments from Reviewer2

Comment: 1. In Figure 1, the authors could add an additional diagram showing the chi and theta angles on a single CCT subunit with twisting and nodding motions to provide more physical intuition.

Reply:

Thanks for the heads up. I have added a diagram to the inset of Figure 1E to visually show which movements the angles of chi and theta represent.

Comment: 2. For Figure 1H, it would be helpful to provide some more descriptions of what the image shows and how it confirms the DXT result for people who are unfamiliar with AFM experiments.

Reply:

As you indicated, the AFM analytical images were performed to visually confirm that the CCT complexes were properly immobilized on the substrate under the experimental conditions implemented for the DXT analysis. The conditions allowed the CCT complexes to be immobilized on the plane properly, which played an important role in pre-determining the optimal density concentration for the DXT analysis. For those who are not familiar with AFM, Revise has modified and added the following explanatory text to the legend of Figure 1 (H).  (I have highlighted the corrected text in red in the revised manuscript.)

"(H) Atomic Force Microscopy (AFM) confirmed immobilization of CCT molecules on the anti-PA tag antibody coated substrate, with No.2 used as a representative example. AFM operates a tiny probe, on a flexible cantilever beam, to interact with a sample and adjust its height based on the cantilever deflection. By scanning the probe across various points of the sample's surface, a detailed nano-scale image of the sample can be generated. Brighter areas indicate higher elevations."

The following text has also been added and revised in the text (Lines 119-123).

To confirm the antibody-mediated immobilization of the CCT complex on the surface, we employed atomic force microscopy (AFM) (Figure 1H). Each round-shaped particle, which represents an individual CCT complex, was properly immobilized on the antibody-coated surface.

Comment: 2. In the discussion and Figure 5, the authors hypothesized that the limited mobility of No.2 can be explained by the role of the intermediate domain (tagged with No.2) as the rotation axis for CCT subunits using normal mode analysis. Another computational study of TRiC/CCT using elastic network analysis of Cryo-EM structures, Yan Zhang et al. Progress in Biophysics and Molecular Biology (2021) 160: 104-120, also showed the role of the intermediate domain as the rotation axis in both the twisting and nodding motion (supplementary movies), which might be relevant for this study.

3. Several other elastic network model (ENM) studies on chaperones and large biomolecular complexes could enrich your discussion. I recommend considering the following publications:

(a) Ivet Bahar et al., Biophysical Journal (2015) 109(6)

(b) Pablon Chacon et al., J. Mol. Bio. (2003), 326

(c) She Zhang et al., Bioinformatics (2021), 37(20)

Reply:

Thank you very much for your information. We have cited Yan Zhang et al. (2021) to support our claim that the intermediate domain could play a role as the rotation axis in both the twisting and nodding motion of CCT subunits. We have also added the recommended ENM studies by Bahar et al. (2015), Chacon et al. (2003), and Zhang et al. (2021) to support the usefulness of ENM studies for investigating the dynamics of large biomolecular complexes. We have also modified the text (Lines 284-285). 

Comment: (a) Label the substrate circle/oval with “substrate” or “actin” instead of describing it in the caption.

 Reply:

As advised, the label "sub" (substrate) was added in the Figure 5. Correspondingly, we have also modified the figure legend.

Comment: (b) Label the PA tags as No.1, No.2, and No.3, consistent with Figure 1 and the texts.

Reply:

As advised, the descriptions of PA tags were revised in Figure 5, consistent with Figure 1 and the texts.

Comment: (c) Perhaps label the light blue (red) as weak positive (negative) in the figure or use legends. Otherwise, it is easy to miss this among the long captions.

Reply:

As advised, we have revised the labels in Figure 5. Light blue indicates a weak negative, and light red indicates a weak positive.

Comment: (d) For the open state, although normal modes are bidirectional, it may be clearer for readers to either change the arrows of the protein structure to show an inward movement, consistent with the other two panels, or point this out in the caption.

Reply:

We would appreciate your pointing this out. We decided to show the movement of the CCT complex more three-dimensionally by showing the Normal mode results shown in the Figure 5 from the OUTER view and SIDE view . Also, as you pointed out, we added arrows to show the direction of movement more clearly, and also added whether the direction of movement represents a twist (χ) or tilt (θ).

We appreciate your time and effort in providing us with valuable feedback.

Reviewer 3 Report

Dear Authors,

Congratulations on the excellent experimental work, but some of my minor comments should be taken into account:

1. The abbreviation “PA-tag” should be deciphered at the first mention (legend to Fig. 1).

2. The reference [13] in the Fig. 1 legend should be as [8].

3. The references [40, 52-54] in the Fig. 5 legend should be before [32], and appropriate changes to the reference numbers should be made.

4. The reference 15 indicated in the list of references (p. 15) seems to be incorrect.

5. Electron microscopy, indicated in “Materials and Methods” as paragraph 4.6, is not mentioned in the text.  

Author Response

Comment: 1. The abbreviation “PA-tag” should be deciphered at the first mention (legend to Fig. 1).

Reply:

The abbreviation "PA-tag" was not clearly defined in the original paper, so we will cite the paper instead in the figure1G legend. We assume that it is similar to the "podoplanin affinity tag" (PA-tag), but we cannot be sure.

(G) The anti-PA tag antibodies [9] were immobilized on a gold- and chromium-coated Kapton surface via thiolate bonds formed by the cysteine residues on the antibodies.”

Comment: 2. The reference [13] in the Fig. 1 legend should be as [8].

Comment: 3. The references [40, 52-54] in the Fig. 5 legend should be before [32], and appropriate changes to the reference numbers should be made.

Comment: 4. The reference 15 indicated in the list of references (p. 15) seems to be incorrect.

Reply:

The above references to the papers were not in the proper order. We have corrected the order appropriately as you indicated. Thank you for pointing this out.

Comment: 5. Electron microscopy, indicated in “Materials and Methods” as paragraph 4.6, is not mentioned in the text.

Reply:

Thank you for pointing this out. Following text regarding the use of electron microscopy was added in the text (Lines 118-119).

The structure of these purified complexes was then verified using electron microscopy, as shown in Supplementary Figure S1.

We appreciate your time and effort in providing us with valuable feedback.

Round 2

Reviewer 1 Report

The manuscript is revised well.